# A Causal Approach to Detecting Multivariate Time-series Anomalies and Root Causes

## Abstract

Detecting anomalies and the corresponding root causes in multivariate time series plays an important role in monitoring the behaviors of various real-world systems, e.g., IT system operations or manufacturing industry. Previous anomaly detection approaches model the joint distribution without considering the underlying mechanism of multivariate time series, making them computationally hungry and hard to identify root causes. In this paper, we formulate the anomaly detection problem from a causal perspective and view anomalies as instances that do not follow the regular causal mechanism to generate the multivariate data. We then propose a causality-based framework for detecting anomalies and root causes. It first learns the causal structure from data and then infers whether an instance is an anomaly relative to the local causal mechanism whose conditional distribution can be directly estimated from data. In light of the modularity property of causal systems (the causal processes to generate different variables are irrelevant modules), the original problem is divided into a series of separate, simpler, and low-dimensional anomaly detection problems so that where an anomaly happens (root causes) can be directly identified. We evaluate our approach with both simulated and public datasets as well as a case study on real-world AIOps applications, showing its efficacy, robustness, and practical feasibility.

## 1 Introduction

Multivariate time series is ubiquitous in monitoring the behavior of complex systems in real-world applications, such as IT operations management, manufacturing industry and cyber security (Hundman et al., 2018; Mathur & Tippenhauer, 2016; Audibert et al., 2020). Such data includes the measurements of the monitored components, e.g., the operational KPI metrics such as CPU/Database usages in an IT system. An important task in managing these complex systems is to detect unexpected observations deviated from normal behaviors, figure out the root causes of abnormal behaviors, and notify the operators timely to resolve the underlying issues. Detecting anomalies and corresponding root causes in multivariate time series aims to accomplish this task and has been actively studied in machine learning, which automate the identification of issues and incidents for improving system availability in AIOps (AI for IT Operations) (Dang et al., 2019).

Various algorithms have been developed to detect anomalies in multivariate time series data. In general, there are two kinds of directions commonly explored, i.e., treating each dimension individually using univariate time series anomaly detection algorithms (Hamilton, 1994; Taylor & Letham, 2018; Ren et al., 2019), and treating all the dimensions as an entity using multivariate time series anomaly detection algorithms (Zong et al., 2018; Park et al., 2017; Su et al., 2019). The first direction ignores the dependencies between different time series, so it may be problematic especially when sudden changes of a certain dimension do not necessarily mean failures of the whole system, or the relations among the time series become anomalous (Zhao et al., 2020). The second direction takes the dependencies into consideration, which are more suitable for real-world applications where the overall status of a system is more concerned about than a single dimension. Recently, deep learning receives much attention in anomaly detection, e.g., DAGMM (Zong et al., 2018), LSTM-VAE (Park et al., 2017) and OmniAnomaly (Su et al., 2019), which infer dependencies between different time series and temporal patterns within one time series implicitly. Recently, Dai & Chen (2022) developed a graph-augmented normalizing flow approach that models the joint distribution via the learned DAG. However, the dependencies inferred by deep learning models do not represent the underlying pro-

cess of generating the observed data and the causal relationships between time series are ignored; such methods do not provide a mechanistic understanding of anomalies and it is hard for them to identify the root causes when an anomaly occurs.

In real-world applications, root cause analysis (RCA) is traditionally treated as a module separated from anomaly detection, identifying potential root causes given the detected anomalous metrics by analyzing the dependencies between the monitored metrics (Soldani & Brogi, 2021). Because RCA requires to know which metric is anomalous, univariate (instead of multivariate) time series anomaly detection algorithms are mostly applied to detect anomalies, and then RCA analyzes system/service graphs obtained via domain knowledge or observed data to determine root causes. Both univariate and multivariate algorithms have drawbacks and cannot be integrated with RCA seamlessly.

To overcome these issues, we take a causal perspective (Pearl, 2009; Spirtes et al., 1993) to naturally view anomalies in multivariate time series as instances that do not follow the regular causal mechanism, and propose **a novel causality-based framework for detecting anomalies and root causes simultaneously**. Specifically, our approach leverages the causal structure discovered from data so that the joint distribution of multivariate time series is factorized into simpler modules where each module corresponds to a local causal mechanism, reflected by the corresponding conditional distribution. Those local mechanisms are modular or autonomous (Pearl, 2009), and can then be handled separately, which is known as the modularity property of causal systems. In light of this property, the problem is then naturally decomposed into a series of low-dimensional anomaly detection problems. Each sub-problem is concerned with a local mechanism. Because we focus on issues with separate local causal mechanisms, the root causes of an anomaly can be identified at the same time. The main contributions of this paper are summarized below.

- We reformulate anomaly detection and root cause analysis of multivariate time series from a causality perspective, which helps understand where and how anomalies happen and facilitates anomaly detection in light of the understanding.

- We propose a novel framework that decomposes the multivariate time series anomaly detection problem into a series of separate low-dimensional anomaly detection problems by exploiting the causal structure discovered from data, which not only detects the anomalies more accurately but also offers a natural way to find their root causes.

- We perform empirical studies of evaluating our approach with both simulation and public datasets as well as a case study of an internal real-world AIOps application, validating its efficacy and robustness to different causal discovery techniques and settings.

Our formulation offers an alternative understanding of anomalies: an anomaly is a data point that does not follow the regular data-generating process. The modularity property makes our approach simpler to train, suitable for real-world applications and easier for root cause analysis. Our method can detect those anomalies that are hard for the approaches based on modeling marginal/joint distributions only, illustrating the benefit of the causal view and treatment of anomalies.

## 2 THE CAUSAL APPROACH

Given a multivariate time series $\mathbf{X}$ with length $T$ and $d$ variables, i.e., $\mathbf{X} = \{\mathbf{x}_1, \mathbf{x}_2, \cdots, \mathbf{x}_d\} \in \mathbb{R}^{T \times d}$, let $x_i(t)$ be the observation of the $i$th variable measured at time $t$. The task in this paper is to detect anomalies after time step $T$ that differ from the regular points in $\mathbf{X}$ significantly and identify the corresponding root causes, i.e., test whether $\mathbf{X}_j$ for $j > T$ follows its regular distribution or not.

### 2.1 WHY THE CAUSAL VIEW MATTERS

Let us consider a simple example shown in Figure 1, i.e., the measurements of three components $x, y, z$ with causal structure $x \rightarrow y \rightarrow z$. An anomaly labeled by a black triangle happens at time step $40$, where the causal mechanism between $x$ and $y$ becomes abnormal. Typically it is hard to find such an anomaly based on the marginal distributions or the joint distribution. But from local causal mechanism $p(y|x)$, such anomaly becomes obvious, e.g., $p(y|x)$ is much lower than its normal values. In this example, at time step $40$ the probability **densities** $p(x) = 0.786$, $p(y) = 1.563$, $p(z) = 1.695$, $p(x, y, z) = p(x)p(y|x)p(z|y) = 0.046$ while $p(y|x) = 0.011$,

meaning that it is easier to find this anomaly by examining the local causal mechanism $p(y|x)$. **A real-world motivation example can be found in Figure 8.**

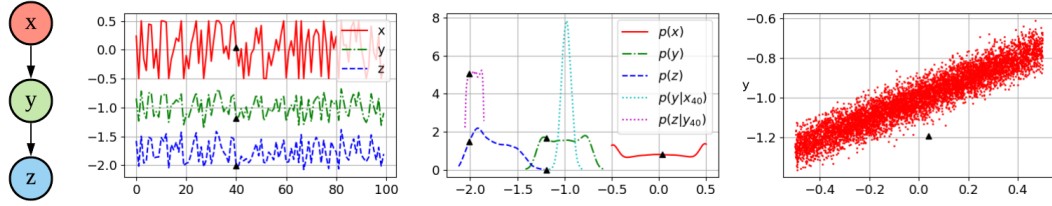

Figure 1: The causal mechanism between $x, y, z$ is $y = 0.5x + \epsilon_1$, $z = \tanh(y^2 - y) + \epsilon_2$. An anomaly occurs at time step 40 labeled by a black triangle. The causal mechanism helps find the anomaly easily as the p-value w.r.t. $y$ conditioned on $x$ is 0.011 (test whether $y$ follows its normal conditional distribution $p(y|x)$). The root cause of this anomaly is $y$ because only $p(y|x)$ is anomalous.

If the causal structure of the underlying process is given, we can examine whether each variable in the time series follows its regular causal mechanism. The causal mechanism can be represented by the structural equation model, i.e., $x_i(t) = f_i(\mathbb{PA}(x_i(t)), \epsilon_i(t)), \forall i = 1, \cdots, d$, where $f_i$ are arbitrary measurable functions, $\epsilon_i(t)$ are independent noises and $\mathbb{PA}(x_i(t))$ represents the causal parents of $x_i(t)$ including both lagged and contemporaneous ones (Pearl, 2009). This causal structure can also be represented by a causal graph $\mathcal{G}$ whose nodes correspond to the variables $x_i(t)$ at different time lags. In this paper, we assume the graph $\mathcal{G}$ is a directed acyclic graph (DAG) and that the causal relationships are stationary unless an anomaly occurs. According to the Markov factorization, the joint distribution of $\mathbf{x}(t)$ can be factored as $\mathbb{P}[\mathbf{x}(t)] = \prod_{i=1}^{d} \mathbb{P}^{\mathcal{G}}[x_i(t)|\mathbb{PA}(x_i(t))]$ where $\mathbb{P}^{\mathcal{G}}$ denotes the conditional distribution.

The local causal mechanisms, corresponding to these conditional distribution terms, are known to be irrelevant to each other in a causal system (Pearl, 2009). An anomaly can then be identified according to the local causal mechanism. Therefore, we define anomalies as follows.

**Definition 1** *A point $\mathbf{x}(t)$ at time step $t$ is an anomaly if there exists at least one variable $x_i$ such that $x_i(t)$ violates the local generating mechanism, i.e., given $\mathbb{PA}(x_i(t))$, $x_i(t)$ does not follow $\mathbb{P}^{\mathcal{G}}[x_i(t)|\mathbb{PA}(x_i(t))]$, which is the conditional distribution corresponding to the regular causal mechanism.*

This definition states that an anomaly happens in the system if the causal mechanism between a variable and its causal parents are violated, e.g., the local causal effect dramatically varies (Fig 1), or a big change happens on a variable and this change propagates to its children. Based on Definition 1, the anomaly detection problem can be divided into several low-dimensional subproblems, e.g., by checking whether each variable follows its regular conditional distribution. Thank to this modularity property, the root causes can be naturally identified when an anomaly event occurs. Here is our definition of root causes.

**Definition 2** *The root causes of an anomaly point $\mathbf{x}(t)$ are those variables $x_i$ such that given $\mathbb{PA}(x_i(t))$, $x_i(t)$ does not follow $\mathbb{P}^{\mathcal{G}}[x_i(t)|\mathbb{PA}(x_i(t))]$, e.g., an anomaly happens on the local causal mechanism related to those variables.*

Definition 2 indicates that $x_i$ is one of the root causes if the local causal mechanism of variable $x_i(t)$ is violated. In Figure 1, variable $y$ is the root cause by our definition because the causal mechanism between $y$ and $z$ is normal while the causal mechanism between $x$ and $y$ is violated.

## 2.2 METHOD

We consider the unsupervised learning setting where $\mathbf{X}$ is given as the training data for learning the graph structures and the conditional distributions. (We will discuss the effects of possible anomalies on the learned causal structure in Section 2.2.4.) For learning causal graphs, we exploit suitable causal discovery methods, as discussed in Section 2.2.1. For learning conditional distributions, we maximize the log likelihoods given the observation data, i.e., maximizing

$L_i(\mathbf{X}) = \sum_{t=1}^{T} \log \mathbb{P}^{\mathcal{G}}[x_i(t)|\mathbb{PA}(x_i(t))], \forall i = 1, \cdots, d$. Specifically, let $\mathcal{C}_{\mathcal{R}}$ be the set of variables with no causal parents in $\mathcal{G}$. There are two cases to be considered:

- Variable with parents ($i \notin \mathcal{C}_{\mathcal{R}}$): The conditional distribution of $x_i(t)$ given its causal parents needs to be estimated, i.e., $\mathbb{P}^{\mathcal{G}}[x_i(t)|\mathbb{PA}(x_i(t))]$ is modeled via conditional density estimation, which can be learned in a supervised manner.

- Variable with no parents ($i \in \mathcal{C}_{\mathcal{R}}$): We model $\mathbb{P}[x_i(t)]$ by applying any existing method for modeling time series with the historical data $H_i(t) = \{x_i(1), \cdots, x_i(t-1)\}$ of $x_i$, meaning that our framework can leverage the state-of-the-art time series models.

The training step produces the causal graph and the estimated conditional distributions corresponding to local causal mechanisms (Section 2.2.2). For anomaly detection of local causal mechanism, we detect data points that do not follow regular conditional distributions. There are multiple possible ways to compute the final anomaly score; Heard & Rubin-Delanchy (2018) compared six methods for combining p-values from individual tests, and showed that taking the minimum is sensitive to the smallest p-value, which is suitable for reporting anomalies that any of the metrics is abnormal. Hence the anomaly score is defined as one minus the minimum value of these estimated probabilities. Intuitively, the purpose of using the minimum function is that we expect the algorithm to report an anomaly if any of the metrics (root variables) or local causal mechanisms (conditional probabilities) becomes abnormal, i.e., a data point is labeled as an anomaly if its anomaly score is larger than a certain threshold. If an anomaly event is detected, the root cause scores are computed for each variable and then the variables with the top scores are selected as the root causes (Section 2.2.3). Algorithm 1 outlines our approach. The anomalies in training data may decrease the performance. We discuss this issue and provide a solution for handling training anomalies in Section 2.2.4.

---

**Algorithm 1** The causality-based approach for detecting anomalies and root causes

---

**Input**: training data $\mathbf{X} = \{\mathbf{x}_i\}_{i=1}^{d} \in \mathbb{R}^{T \times d}$, test data $\mathbf{Y} = \{\mathbf{y}_i\}_{i=1}^{d} \in \mathbb{R}^{\hat{T} \times d}$, and threshold $\lambda$;

**Training procedure**:

1: Infer the causal graph $\mathcal{G}$ via causal discovery techniques, e.g., FGES (Chickering, 2003; 2002; Meek, 1995) and PC (Spirtes & Glymour, 1991). If the $\mathcal{G}$ is a partial DAG, convert it into a DAG by the method (Dor & Tarsi, 1992); (Section 2.2.1)
2: For variable $i$, train a model $\mathcal{M}_i$ estimating conditional distribution $\mathbb{P}^{\mathcal{G}}[x_i(t)|\mathbb{PA}(x_i(t))]$ with training data $\{x_i(t), \mathbb{PA}(x_i(t))\}_{t=1}^{T}$, where $\mathbb{PA}(x_i(t))$ can be an empty set. (Section 2.2.2)

**Detection procedure:**

1: **for** $t = 1$ to $\hat{T}$ **do**
2:    Compute anomaly score: $\mathbb{A}(\mathbf{y}(t)) = 1 - \min\{\mathcal{M}_i(y_i(t))|i = 1, \cdots, d\}$;
3:    Set anomaly label $l_t = 1$ if $\mathbb{A}(\mathbf{y}(t)) > \lambda$ or 0 otherwise;
4:    If anomaly label $l_t$ is 1, computes root cause scores $\mathbb{RS}(x_i(t))$ via Eq (1) for each variable $i$, and set root causes $\mathcal{R}_t$ be the variables with the top-k root cause scores. (Section 2.2.3)
5: **end for**

---

### 2.2.1 CAUSAL DISCOVERY

Our approach needs to exploit the causal structure underlying the data. A traditional way to find causal relations is to use interventions or randomized experiments, which are generally too expensive and time-consuming. Discovering causal information by analyzing purely observational data, known as causal discovery, is then an important problem (Spirtes & Glymour, 1991; Peters et al., 2017; Spirtes & Zhang, 2016). Multiple algorithms have been developed for causal discovery from independent and identically distributed (i.i.d.) or time series data, and their results are asymptotically guaranteed under corresponding assumptions. In this paper, we choose causal discovery algorithms such as PC (Spirtes & Glymour, 1991), FGES (Chickering, 2003; 2002; Meek, 1995), depending on whether we are given temporal data (with time-delayed causal relations) and whether the causal relations are linear or nonlinear. For example, we apply FGES with SEM-BIC score if the variables are linearly related and apply FGES with generalized score function (Huang et al., 2018) if they are non-linearly correlated. One concern is whether the missing or incorrect causal links in the inferred

causal graph have a big impact on the performance of our approach. We performed an empirical study of this impact with public datasets, which shows that interestingly, our approach is robust to the inferred causal graph. The complexity of PC and GES highly depends on the density of the causal graph. Specifically, FGES is highly scalable when dealing with linear models (Ramsey et al., 2016). In real-world applications, e.g., the public datasets in the experiments, even though the variables may not be exactly linearly correlated, FGES can still generate reasonable causal graphs that are good enough for our approach.

### 2.2.2 ANOMALY DETECTION

After the causal Markov factorization, it becomes easier to model the joint distribution compared to the previous approaches, e.g., the conditional distributions representing local causal mechanisms can be estimated using simpler ML models.

For modeling $\mathbb{P}^{\mathcal{G}}[x_i(t)|\mathbb{PA}(x_i(t))]$, one can apply kernel conditional density estimation (Hastie et al., 2009), conditional VAE (CVAE) (Sohn et al., 2015) or even prediction models such as MLP or CNN (Binkowski et al., 2018). Let $\tau_j$ be the causal time lag for a parent $x_j$ and $\tau^*$ be the maximum time lag in $\mathcal{G}$; then we define $\mathbb{PA}^*(x_i(t)) = \{x_j(t-\tau^*), \cdots, x_j(t-\tau_j) \mid j \in \mathbb{PA}\}$. Time lag $\tau_j = 0$ if $x_j$ is a contemporaneous causal parent of $x_i$. For causal parent $x_j$, more of its historical data can also be included, e.g., a window with size $k$: $\{x_j(t-\tau_j-k+1), \cdots, x_j(t-\tau_j) \mid j \in \mathbb{PA}\}$. Therefore, the problem becomes estimating the conditional distribution from the empirical observations $\{(x_i(t), c_i(t))\}_{t=1}^T$ where $c_i(t) = \mathbb{PA}^*(x_i(t))$. In this paper, we apply CVAE to model such conditional distribution. The reason why choosing CVAE is that it can be trained fast with a simple architecture and achieve good performance as shown in our experiments. The empirical variational lower bound of CVAE is

$$L(x, c; \theta, \phi) = \frac{1}{n} \sum_{k=1}^{n} \log p_\theta(x|c, z_k) - KL(q_\phi(z|x, c) \parallel p_\theta(z|c)),$$

where $q_\phi(z|x, c)$, $p_\theta(x|c, z_k)$ are MLPs and $p_\theta(z|c)$ is a Gaussian distribution. Given $(x_i(t), c_i(t))$, CVAE outputs $\hat{x}_i(t)$ – reconstruction of $x_i(t)$, and then $\mathbb{P}[x_i(t)|c_i(t)]$ is measured by the distribution of $|\hat{x}_i(t) - x_i(t)|$. [1]

If $\mathbb{PA}(x_i(t))$ is empty, i.e., $i \in \mathcal{C}_\mathcal{R}$, one way to estimate distribution $\mathbb{P}[x_i(t)]$ is to handle $x_i(t)$ via univariate time series models, e.g., ARIMA (Hamilton, 1994), SARIMA (Hyndman & Athanasopoulos, 2018). The other way is to handle the variables in $\mathcal{C}_\mathcal{R}$ together by utilizing the models for multivariate time series anomaly detection, e.g., Isolation Forest (IF) (Liu et al., 2008), AE (Baldi, 2012), LSTM-VAE (Park et al., 2017). The training data for such models includes all the observations of the variables in $\mathcal{C}_\mathcal{R}$, i.e., $\{x_i(t)|i \in \mathcal{C}_\mathcal{R}\}_{t=1}^T$. For example, the training data for a forecasting based method is $\{(x_i(t), \{x_i(t-k), \cdots, x_i(t-1)\})|i \in \mathcal{C}_\mathcal{R}\}_{t=1}^T$ where $x_i(t)$ is predicted by a window of its previous data points.

Our approach reduces to the previous univariate/multivariate time series AD approaches if the causal graph is empty, i.e., no causal relationships are considered. When the causal relationships are available obtained by domain knowledge or data-driven causal discovery techniques, our approach can easily utilize such information and reduces the efforts in joint distribution estimation.

### 2.2.3 ROOT CAUSE ANALYSIS

Root cause analysis (RCA) aims to identify root causes when an anomaly event happens. RCA in real-world applications such as AIOps can be very challenging. One practical issue for identifying root causes is that an anomaly occurs in a variable often makes its causal children variables abnormal due to anomaly propagation. Specifically, based on Definition 2, we propose the following practical algorithm. For variable $x_i$, define its initial root cause score at time $t$ by $\mathbb{S}(x_i(t)) = 1 - \mathcal{M}_i(x_i(t))$. Suppose that $\mathcal{N}(x_i(t))$ is the set of the causal children of $x_i(t)$, the final root cause score is define in a PageRank (Page et al., 1999; Wu et al., 2020) way:

$$\mathbb{RS}(x_i(t)) = \mathbb{S}(x_i(t)) + \alpha \frac{1}{|\mathcal{N}(x_i)|} \sum_{x_j(t) \in \mathcal{N}(x_i)} \mathbb{RS}(x_j(t)), \; \forall i = 1, \cdots, d, \tag{1}$$

---

[1] We assume the reconstruction error is additive, e.g., $x = f(c) + e$, so that $\mathbb{P}(x|c) = \mathbb{P}_e(x - f(c))$. Hence we use the distribution of the reconstruction error for detecting anomalies.

where $\alpha$ is a weight parameter satisfying $0 \leq \alpha < 1$. When $\mathcal{N}(x_i)$ is empty, we set $\mathbb{RS}(x_i(t)) = \mathbb{S}(x_i(t))$. Here the final root cause score of a variable is the weighted combination of its initial root cause score and the final root cause scores of its children to handle the anomaly propagation issue. The root causes at time $t$ are identified by picking the variables with top $\mathbb{RS}$ scores.

### 2.2.4 Negative effect of training anomalies

The existence of anomalies in the training data may decrease the detection performance. Our empirical results show that this issue does not affect the anomaly detection performance much, which is expected to be the case when there are relatively few anomalies in the data. Typically, there are two possible cases where anomalies in training data may have obvious negative impacts on performance. One case is that the value of a metric at certain timestamps becomes extremely large, which can affect the conditional probability estimation. In this case, one can simply remove those values based on statistical rules, e.g., removing them if the absolute value is larger than some threshold in the preprocessing step. The other case is that the proportion of anomalies is relatively large. In this case, we can consider an iterative solution that iteratively updates the causal graph and anomaly detection model, i.e., 1) estimate causal graph $\mathcal{G}$ and train models $\mathcal{M}_i$ with the training data, and 2) remove the anomalies detected by $\mathcal{M}_i$ in the training data and then go to Step (1). We repeat the above two steps until the estimated causal model (including the estimated causal structure and quantitative model, e.g., causal coefficients in the linear case) converges. We conducted an experiment with a simulation dataset to empirically study this iterative solution (Section A.7).

## 3 Experiments

This section evaluates the performance of our proposed approach and compares it to several other approaches. The experiments include: 1) evaluating our approach with simulation and public datasets, 2) analyzing how different causal graphs affect the performance, and 3) a case study demonstrating the application of our approach for real-world anomaly detection in AIOps.

The anomaly detection performance is assessed by the precision, recall and F1-score metrics in a point-adjust manner, i.e., all the anomalies of an anomalous segment are considered as correctly detected if at least one anomaly of this segment is correctly detected while the anomalies outside the ground truth anomaly segment are treated as normal. By default, we apply FGES (Chickering, 2003) to discover the causal graph. For $i \notin \mathcal{C}_{\mathcal{R}}$, we choose CVAE (Sohn et al., 2015). For $i \in \mathcal{C}_{\mathcal{R}}$, we tested the univariate model and other methods such as IF (Liu et al., 2008), AE (Baldi, 2012), LSTM-VAE (Park et al., 2017) in our experiments. We compare our approach with several unsupervised approaches, e.g., AE (Baldi, 2012), DAGMM (Zong et al., 2018), OmniAnomaly (Su et al., 2019), USAD (Audibert et al., 2020), GANF (Dai & Chen, 2022)[2].

### 3.1 Simulation datasets

Section A.3 discusses how to generate simulation datasets. We consider linear/nonlinear causal relationships and three types of anomalies. The first type is a "measurement" anomaly where the causal mechanism is normal but the observation is abnormal due to measurement errors, i.e., randomly pick a node $x_i$, a time step $t$ and a scale $s$, and then set $x_i(t) = [x_i(t) - \mathrm{median}(x_i)] * s + \mathrm{median}(x_i)$. The second type is an "intervention" anomaly, i.e., after generating anomalies for some nodes, those anomaly values propagate to the children nodes according to the causal relationships. The third type is an "effect" anomaly where anomalies only happen on the nodes with no causal children.

**Performance comparison.** In the experiments, we consider six settings derived from the combinations of "linear/nonlinear" and "measurement/intervention/effect". The simulated time series has 15 variables with length 20000, where the first half is the training data and the rest is the test data. The percentage of anomalies is 10%. Table 1 shows the performance of different unsupervised multivariate time series anomaly detection methods with the generated simulated dataset. Clearly, our method outperforms all the other methods. It achieves significantly better F1 scores when the relationships are nonlinear or the anomaly type is "intervention", e.g., ours obtains F1 score 0.759 for "nonlinear, intervention", while the best F1 score achieved by the others is 0.589. In "linear,

---

[2]https://github.com/EnyanDai/GANF

Table 1: Performance comparison (F1-scores) on the simulation datasets.

| | Lin./Measu. | Lin./Inter. | Lin./Effect | Nonlin./Measu. | Nonlin./Inter. | Nonlin./Effect |
|---|---|---|---|---|---|---|
| IF | 0.374 | 0.403 | 0.220 | 0.336 | 0.422 | 0.367 |
| AE | 0.386 | 0.359 | 0.240 | 0.392 | 0.390 | 0.363 |
| VAE | 0.343 | 0.328 | 0.208 | 0.396 | 0.377 | 0.306 |
| LSTM-VAE | 0.457 | 0.454 | 0.485 | 0.581 | 0.545 | 0.393 |
| DAGMM | 0.746 | 0.542 | 0.721 | 0.456 | 0.589 | 0.359 |
| USAD | 0.252 | 0.260 | 0.220 | 0.346 | 0.302 | 0.279 |
| GANF | 0.292 | 0.340 | 0.213 | 0.355 | 0.316 | 0.286 |
| Ours | **0.791** | **0.757** | **0.740** | **0.757** | **0.759** | **0.637** |

measurement/effect", DAGMM has a similar performance with ours because the data can be modeled well by applying dimension reduction followed by a Gaussian mixture model. But when the relationships become nonlinear, it becomes harder for DAGMM to model the data. This experiment shows that the causal mechanism plays an important role in anomaly detection. Modeling joint distribution without considering causality can lead to a significant performance drop.

We use the same simulation datasets as anomaly detection to evaluate the RCA performance measured by the top-k hit ratio, i.e., the predicted top-k root causes are correct as long as one of them is the groundtruth root cause. Table 2 shows the RCA performance of our approach and the baseline. The baseline ignores the causal relationships while samples root causes based on the probabilities proportional to the anomaly scores. Our approach achieves HR@3 $>= 0.95$ and HR@3 $>= 0.75$

Table 2: RCA performance comparison (top-k hit-ratio) on the simulation datasets.

| | Top 1 | Top 2 | Top 3 | Top 4 |
|---|---|---|---|---|
| Linear/Measu. (Baseline) | $0.406 \pm 0.182$ | $0.573 \pm 0.182$ | $0.617 \pm 0.160$ | $0.631 \pm 0.148$ |
| Linear/Measu. (Ours) | $0.654 \pm 0.117$ | $0.916 \pm 0.075$ | $0.965 \pm 0.040$ | $0.993 \pm 0.010$ |
| Linear/Inter. (Baseline) | $0.463 \pm 0.182$ | $0.551 \pm 0.186$ | $0.596 \pm 0.171$ | $0.614 \pm 0.163$ |
| Linear/Inter. (Ours) | $0.637 \pm 0.178$ | $0.815 \pm 0.126$ | $0.960 \pm 0.032$ | $0.988 \pm 0.020$ |
| Nonlinear/Measu. (Baseline) | $0.253 \pm 0.042$ | $0.449 \pm 0.040$ | $0.571 \pm 0.074$ | $0.644 \pm 0.049$ |
| Nonlinear/Measu. (Ours) | $0.577 \pm 0.081$ | $0.656 \pm 0.056$ | $0.764 \pm 0.091$ | $0.847 \pm 0.088$ |
| Nonlinear/Inter. (Baseline) | $0.262 \pm 0.048$ | $0.439 \pm 0.113$ | $0.589 \pm 0.114$ | $0.635 \pm 0.102$ |
| Nonlinear/Inter. (Ours) | $0.541 \pm 0.152$ | $0.623 \pm 0.115$ | $0.776 \pm 0.077$ | $0.867 \pm 0.118$ |

for the "linear" and "nonlinear" settings, respectively, which is significantly better than the baseline.

## 3.2 PUBLIC DATASETS

Four public datasets were used in our experiments: 1) Secure Water Treatment (SWaT) (Mathur & Tippenhauer, 2016): it consists of 11 days of continuous operation, i.e., 7 days collected under normal operations and 4 days collected with attacks, 2) Water Distribution (WADI) (Mathur & Tippenhauer, 2016): It consists of 16 days of continuous operation, of which 14 days were collected under normal operation and 2 days with attacks. 3) Soil Moisture Active Passive (SMAP) satellite and Mars Science Laboratory (MSL) rover Datasets (Hundman et al., 2018), which are two public datasets expert-labeled by NASA.

**Performance comparison.** Table 3 shows the results on four representative datasets. Overall, IF, AE, VAE and DAGMM have relatively lower performance because they neither exploit the temporal information nor leverage the causal relationships between those variables. LSTM-VAE, Omni-Anomaly and USAD perform better than these four methods since they utilize the temporal information via modeling the current observations with the historical data, while the DAG-based method GANF does not perform well except for SWaT. Our approach exploits the causal relationships besides the temporal information, achieving significantly better results than the other methods in all the datasets, e.g., ours has the best F1 score 0.918 for SWaT and 0.818 for WADI, while the best F1 scores for SWaT and WADI by other methods are 0.846 and 0.767, respectively. For each public datasets, Table 4 reports the best metrics that can be achieved by choosing the best thresholds in the test datasets. Clearly, if we are allowed to choose better thresholds, the metrics achieved by our

Table 3: Performance comparison of our approach and other methods on the public datasets.

| Methods | SMAP | | | MSL | | | SWaT | | | WADI | | |
|---|---|---|---|---|---|---|---|---|---|---|---|---|
| | Prec. | Recall | F1 | Prec. | Recall | F1 | Prec. | Recall | F1 | Prec. | Recall | F1 |
| IF | 0.815 | 0.591 | 0.685 | 0.854 | 0.922 | 0.887 | 0.998 | 0.669 | 0.801 | 0.541 | 0.794 | 0.644 |
| AE | 0.806 | 0.585 | 0.678 | 0.858 | 0.892 | 0.875 | 0.999 | 0.656 | 0.792 | 0.595 | 0.762 | 0.668 |
| VAE | 0.808 | 0.588 | 0.681 | 0.771 | 0.656 | 0.709 | 0.999 | 0.656 | 0.792 | 0.616 | 0.855 | 0.716 |
| LSTM-VAE | 0.818 | 0.591 | 0.686 | 0.859 | 0.911 | 0.884 | 0.997 | 0.689 | 0.815 | 0.658 | 0.920 | 0.767 |
| DAGMM | 0.800 | 0.877 | 0.837 | 0.900 | 0.864 | 0.882 | 0.829 | 0.767 | 0.797 | 0.639 | 0.501 | 0.412 |
| OmniAnom | 0.758 | 0.975 | 0.853 | 0.901 | 0.889 | 0.895 | 0.722 | 0.983 | 0.833 | 0.265 | 0.980 | 0.417 |
| USAD | 0.769 | 0.983 | 0.863 | 0.861 | 0.964 | 0.910 | 0.987 | 0.740 | 0.846 | 0.645 | 0.322 | 0.430 |
| GANF | 0.692 | 0.549 | 0.612 | 0.285 | 0.773 | 0.416 | 0.964 | 0.706 | 0.815 | 0.576 | 0.596 | 0.586 |
| Ours | 0.874 | 0.982 | **0.925** | 0.867 | 0.961 | **0.912** | 0.945 | 0.892 | **0.918** | 0.749 | 0.901 | **0.818** |
| (std) | ±0.001 | ±0.006 | ±0.003 | ±0.003 | ±0.011 | ±0.007 | ±0.009 | ±0.016 | ±0.008 | ±0.021 | ±0.029 | ±0.023 |

approach can be much higher, e.g., F1-score 0.946 for SMAP and 0.913 for MSL. We also report the running time of our approach in Section A.8.

Table 4: The best performance of our approach with the public datasets.

| Dataset | SMAP | MSL | SWaT | WADI |
|---|---|---|---|---|
| Precision* | $0.951 \pm 0.011$ | $0.903 \pm 0.029$ | $0.929 \pm 0.018$ | $0.883 \pm 0.021$ |
| Recall* | $0.930 \pm 0.011$ | $0.951 \pm 0.033$ | $0.965 \pm 0.009$ | $0.947 \pm 0.020$ |
| F1* | $0.940 \pm 0.004$ | $0.926 \pm 0.017$ | $0.946 \pm 0.007$ | $0.913 \pm 0.008$ |

**Ablation study on $\mathcal{M}_i$ for $i \in \mathcal{C}_\mathcal{R}$.** This experiment evaluates the effect of the causal information on anomaly detection. For an anomaly detection method $\mathcal{A}$ such as IF and AE, we compare $\mathcal{A}$ with our approach "ours + $\mathcal{A}$" that uses CVAE for $i \notin \mathcal{C}_\mathcal{R}$ (estimates $\mathbb{P}^\mathcal{G}[x_i(t)|\mathbb{P}\mathbb{A}(x_i(t))]$) and $\mathcal{A}$ for $i \in \mathcal{C}_\mathcal{R}$ (estimates $\prod_{i \in \mathcal{C}_\mathcal{R}} \mathbb{P}[x_i(t)]$). We report the metrics as mentioned above and the best metrics achieved by choosing the best thresholds in the test datasets. Table 5 shows the performance of

Table 5: Performance of our method using different models for $\mathcal{A}$ in SWaT and WADI. "*" means the best metrics. $\mathcal{A} = \emptyset$ means anomalies are detected by $i \notin \mathcal{C}_\mathcal{R}$ only without using $i \in \mathcal{C}_\mathcal{R}$.

| $\mathcal{A}$ | SWaT | | | | | | WADI | | | | | |
|---|---|---|---|---|---|---|---|---|---|---|---|---|
| | Prec. | Rec. | F1 | Prec.* | Rec.* | F1* | Prec. | Rec. | F1 | Prec.* | Rec.* | F1* |
| $\emptyset$ | 0.952 | 0.874 | 0.911 | 0.950 | 0.929 | 0.940 | 0.749 | 0.920 | 0.826 | 0.873 | 0.979 | 0.923 |
| IF | 0.947 | 0.893 | 0.919 | 0.946 | 0.945 | 0.945 | 0.738 | 0.920 | 0.819 | 0.948 | 0.920 | 0.934 |
| AE | 0.958 | 0.900 | 0.928 | 0.963 | 0.920 | 0.941 | 0.789 | 0.920 | 0.850 | 0.931 | 0.979 | 0.955 |
| LSTM-VAE | 0.954 | 0.874 | 0.912 | 0.951 | 0.936 | 0.944 | 0.748 | 0.920 | 0.825 | 0.949 | 0.920 | 0.934 |

our approach with different $\mathcal{A}$, where $\mathcal{A} = \emptyset$ means that the anomalies are detected by $i \notin \mathcal{C}_\mathcal{R}$ only without using $i \in \mathcal{C}_\mathcal{R}$. By comparing this table with Table 3 we can observe that "ours + $\mathcal{A}$" performs much better than using $\mathcal{A}$ only, e.g., "ours + AE" achieves F1 score 0.850 for WADI, while AE obtains 0.668 for WADI. If $\mathcal{A}$ is not used in anomaly detection, we get a performance drop in terms of F1 score. For example, the best F1 score drops from 0.934 to 0.923 for WADI.

**Ablation study on causal discovery and causal mechanism estimation.** We also studied the effects of different parameters for discovering causal graphs on the performance of our approach. The experiments (in Section A.9) shows that our approach is robust to the changes of the inferred causal graph. In practice, the causal graph is not required to be accurate, namely, we just need to ensure that it doesn't contain too many missing links or false positive links. Besides FGES, other methods such as the PC algorithm (Spirtes & Glymour, 1991) can also be applied to infer the causal graphs. The causal graphs inferred by PC are probably different from those computed by FGES. Our experiments show that our anomaly detection approach is stable even though the causal graphs are different. Table 6 compares the performance of our approach with FGES, GES and PC. For SWaT, using FGES, GES and PC have similar performance. For WADI, using PC performs worse than using GES and FGES, but the F1-score 0.768 is still better than the other approaches. The performance drop is because the causal graph discovered by FGES is more accurate than PC in

Table 6: Performance comparison (F1-score) with the public datasets, e.g., GES vs PC vs FGES for causal discovery, and CVAE vs MLP vs KMN for causal mechanism estimation.

| Methods | SWaT | WADI | Methods | SWaT | WADI |
|---|---|---|---|---|---|
| Ours (GES) | $0.895 \pm 0.013$ | $0.802 \pm 0.015$ | Ours (CVAE) | $0.918 \pm 0.008$ | $0.818 \pm 0.023$ |
| Ours (PC) | $0.912 \pm 0.007$ | $0.768 \pm 0.020$ | Ours (MLP) | $0.909 \pm 0.010$ | $0.739 \pm 0.041$ |
| Ours (FGES) | $0.918 \pm 0.008$ | $0.818 \pm 0.023$ | Ours (KMN) | $0.900 \pm 0.007$ | $0.634 \pm 0.029$ |

WADI. As shown in Table 6, we also tested different methods for estimating causal mechanisms (conditional distributions), e.g., CVAE, MLP and KMN (Ambrogioni et al., 2017). CVAE works better than the others in SWaT and WADI so we choose CVAE by default.

### 3.3 CASE STUDY: REAL-WORLD APPLICATION IN AIOPS

Our last experiment is to apply our method for a real-world anomaly detection task in AIOps, where the goal is to monitor the operational key performance indicator (KPI) metrics of database services for alerting anomalies and identifying root causes in order to automate remediation strategies and improve database availability in cloud-based services. In this application, we monitor a total of 61 time series variables measuring the KPI metrics of database services, e.g., read/write IO requests, CPU usage, DB time. The data in this case study consists of the one-month measurements. According to the feedback from domain experts, most of the inferred causal relationships shown in Figure 2 are consistent with the known domain knowledge. For example, the discovered links Redo (redo size) –> Lfpw (log file parallel write) –> Lfs (log file sync) –> COMT (commit) are exactly the same as the domain knowledge.

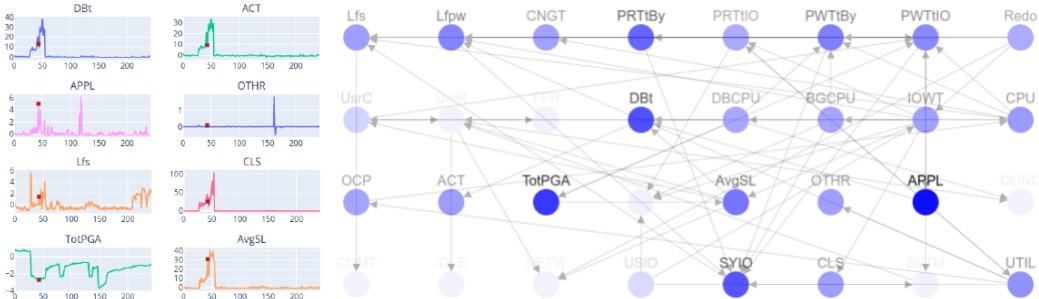

Figure 2: Example on a real-world AIOps case, showing 8 out of 61 time variables (left) and a part of the causal graph (right). The deeper colors of nodes in the graph indicate higher root cause scores.

The incidences that happened are relatively rare, e.g., 2 major incidences one month, and our anomaly detection approach correctly detect these incidences. Therefore, we focus on the root cause analysis in this case study. Figure 2 shows an example of one major incidence, showing several abnormal metrics such as DBt (DB time), Lfs (log file sync), APPL (application), TotPGA (total PGA allocated) and a part of the causal graph. The root cause scores computed by our method are highlighted. We can observe that the top root causes metrics are APPL, DBt and TotPGA, all of which correspond to application or database related issues for the incident as validated by domain experts. More results can be found in Appendix.

### 4 CONCLUSIONS

Most previous approaches for multivariate time series anomaly detection model the joint distribution directly without considering the underlying causal process of the observed time series data. This paper presented a new definition and formulation of anomalies in multivariate time series from a causal perspective, and proposed a novel approach that exploits the causal structures discovered from data to help detect anomalies more accurately and identify the root causes robustly according to the local causal mechanism. Our experiments on both simulation and real datasets demonstrated the efficacy, robustness and practical feasibility of the proposed approach in real-world applications.

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

# A  APPENDIX

## A.1  RELATED WORK

Anomaly detection methods for univariate time series can be applied to each dimension of multivariate time series. Popular univariate anomaly detection techniques include statistical or distance-based methods, e.g., KNN (Chaovalitwongse et al., 2007; Angiulli & Pizzuti, 2002), One-Class SVM (Manevitz & Yousef, 2002), and probabilistic methods (Chandola et al., 2009). These methods are computationally efficient and suitable for high dimensional data. But their performance degrades faced with long-term anomalies since the temporal patterns within time series are ignored. To address this issue, temporal prediction methods, e.g., ARIMA, SARIMA (Hamilton, 1994), Prophet (Taylor & Letham, 2018), SR-CNN (Ren et al., 2019), and DONUT (Laptev et al., 2015), have been proposed to model temporal dependencies/autocorrelations. However, these methods treat each dimension individually and ignore the correlations between different time series. As shown in Figure 1, they cannot identify the anomaly corresponding to the abnormal causal mechanism.

Recent years have seen the increasing popularity of unsupervised methods using deep learning techniques, which can infer the correlations between time series. For example, DAGMM (Zong et al., 2018) combines an autoencoder with a Gaussian mixture model to model the joint distribution. MSCRED (Zhang et al., 2019) utilizes the system signature matrix to model the correlations and temporal patterns. LSTM-VAE (Park et al., 2017) combines LSTM with VAE and models temporal

dependencies through LSTM. OmniAnomaly (Su et al., 2019) learns robust time series representations with a stochastic variable connection and a planar normalizing flow. USAD (Audibert et al., 2020) uses adversely trained autoencoders inspired by GANs, providing fast training. However, these methods model the joint distribution directly without considering the process behind multivariate time series, and an anomaly that happens to a local mechanism in the process might not change the joint distribution dramatically. Besides, it is difficult for them to leverage the domain knowledge of the monitored system, e.g., the known causal dependencies between time series, and to provide explanations that are crucial for root cause analysis and remediation when an anomaly occurs. Finally, our work also differs substantially from existing studies (Qiu et al., 2012; 2020) which though explore causality in anomaly detection in different ways, but do not use the causal mechanism to model anomalies in time series.

Root cause analysis (RCA) methods leverage the KPI metrics monitored on those services to determine the root causes when an anomaly event is detected. The key idea behind RCA with KPI metrics is to analyze the relationships or dependencies between these metrics and then utilize these relationships to identify root causes when an anomaly occurs. Typically, there are two types of approaches: 1) identifying the anomalous metrics in parallel with the observed anomaly via metric data analysis, and 2) discovering topology/causal graphs that represent the causal relationships between the services.

Nguyen et al. (2011; 2013) propose two similar RCA methods by analyzing low-level system metrics, e.g., CPU, memory and network statistics. Both methods first detect abnormal behaviors for each component via a change point detection algorithm when a performance anomaly is detected, and then determine the root causes based on the propagation patterns obtained by sorting all critical change points in a chronological order. Shan et al. (2019) developed a low-cost RCA method called $\epsilon$-Diagnosis to detect root causes of small-window long-tail latency for web services. $\epsilon$-Diagnosis assumes that the root cause metrics of an abnormal service have significantly changes between the abnormal and normal periods. But these methods don't consider the causal relationships between KPI metrics or the dependencies between services in an application.

The second type of RCA approaches leverages such dependencies, which usually involves two steps, i.e., constructing topology/causal graphs given the KPI metrics and domain knowledge, and extracting anomalous subgraphs or paths given the observed anomalies. Such graphs can either be reconstructed from the topology (domain knowledge) of a certain application (Thalheim et al., 2017; Wu et al., 2020; Álvaro Brandón et al., 2020; Samir & Pahl, 2019) or automatically estimated from the metrics via causal discovery techniques (Wang et al., 2018; Mariani et al., 2018; Chen et al., 2019; Meng et al., 2020; Lin et al., 2018; Ma et al., 2019; 2020). To identify the root causes of the observed anomalies, random walk (e.g., Kim et al. (2013); Meng et al. (2020); Wang et al. (2018)), page-rank (e.g., Wu et al. (2020)) or other analysis methods can be applied over the discovered topology/causal graphs. Recently, Budhathoki et al. (2022) proposed a method based counterfactual analysis which identifies the root cause of a detected anomaly/outlier by computing the contribution of each noise term to the anomaly score. But these methods only accept univariate time series anomaly detectors, i.e., detecting anomalies for each metric separately.

## A.2 EXPERIMENTAL SETUP AND PARAMETERS SETTINGS

For the implementation of our approach, we employ the CVAE to model conditional distributions $\mathbb{P}^{\mathcal{G}}[x_i(t)|\mathbb{PA}(x_i(t))]$. We choose the same parameters for all the experiments on both simulated and public real datasets. The encoder and decoder in CVAE are both MLPs with hidden sizes $[10, 20, 10]$. The latent size is 5 and the prior distribution $p_\theta(z|c)$ is assumed to be the standard normal distribution (doesn't depend on $c$). For training CVAE, the optimizer is ADAM with learning rate 0.001, batch size 1024 and epoch num 80.

For modeling $\prod_{i \in \mathcal{R}} \mathbb{P}[x_i(t)]$, there are several options to choose in practical applications. For the simulated datasets and our internal AIOps dataset, we choose a univariate anomaly detection method based on a CNN forecasting model. The CNN forecasting model consists of 4 residual blocks with 1D convolutional layers, i.e., the "(input channels, output channels)" pairs are $(1, 8), (8, 16), (16, 32), (32, 64)$, followed by the concatenate of 1D adaptive average pooling and 1D adaptive max pooling. The output layer is a linear layer. For each residual block, it has two convolutional layers "(input channels, output channels) $->$ (output channels, output channels)". We

choose ADAM as the optimizer with learning rate 0.001 (with decay), batch size 1024 and epoch num 50. The window size of the historical data for prediction is 20. For the public datasets, besides this CNN forecasting model, we can also choose isolation forest (IF) and autoencoder (AE). For IF, the max number of samples is 10000. For AE, the hidden sizes of the encoder are $[25, 10, 5]$, the latent size is 5, and the hidden sizes of the decoder are $[5, 10, 25]$.

For the simulated datasets, we apply FGES and set "max degree = 5" and "penalty discount = 20". For the public datasets SWaT and WADI, we apply FGES and set "max degree = 10" and "penalty discount = 100". For SMAP and MSL, we apply the PC algorithm with the default parameters. The library for causal discovery we used in this project is Tetrad [3]. Smaller "max degree" or larger "penalty discount" in FGES leads to more sparse graphs with less edges. Table 7 lists the number of the edges in the causal graphs discovered with different parameters.

The reason why we choose these parameters such as CVAE hidden sizes = [10, 20, 10] is as follows. For all the simulation datasets, the "max-degree" is set to 5 in FGES and the causal relations are instantaneous, meaning that the number of causal parents of each variable is at most 5 so that the input dimensions of the parent variables in CVAEs for modeling conditional probabilities are at most 5. For the public datasets, the "max-degree" is 10 and we found that there are instantaneous causal influences but not time-delayed ones, so the input dimensions of the parent variables in CVAEs are at most 10. That's why we choose those parameters for the encoder and decoder. For a new dataset, if one considers a similar setting for causal discovery, he/she can use our parameters as default. In general, the input dimensions of CVAEs are at most "max-degree" * "time-lag", so one can choose the hidden sizes around this number. For modeling conditional probabilities, one can construct a validation set by splitting the training dataset. Under the Gaussian distribution assumption in CVAE, the overfitting issue can be found and avoided by measuring the reconstruction MSE loss.

For all the experiments, **the detection thresholds** are inferred by taking the $n$th percentile of the detection scores in the test data, e.g., we choose $n = 95$ for SWaT and WADI, $n = 98$ for SMAP and MSL. For the other methods (except ours) in the simulated datasets, the reported precision, recall and F1-score metrics are the best metrics that can be achieved in the test datasets (by choosing the best threshold).

Table 7: The number of the edges in the causal graphs generated by FGES with different parameters.

| SWaT | max degree (penalty discount = 100) | | | | | | penalty discount (max degree = 10) | | | | | |
|---|---|---|---|---|---|---|---|---|---|---|---|---|
| | d=5 | 6 | 7 | 8 | 9 | 10 | p=20 | 40 | 60 | 80 | 100 | 120 |
| Edge num | 70 | 79 | 88 | 95 | 98 | 102 | 139 | 122 | 115 | 111 | 102 | 93 |
| WADI | max degree (penalty discount = 100) | | | | | | penalty discount (max degree = 10) | | | | | |
| | d=5 | 6 | 7 | 8 | 9 | 10 | p=20 | 40 | 60 | 80 | 100 | 120 |
| Edge num | 152 | 180 | 195 | 211 | 227 | 249 | 331 | 308 | 278 | 262 | 249 | 225 |

### A.3 SIMULATION DATASET

The simulated time series data can be generated in the following steps:

1. Generate an Erdös Rényi random graph $\mathcal{G}$ with number of nodes/variables $n$ and edge creation probability $p$, then convert it into a DAG. We choose $p = 0.1$.

2. For the variables with no parents in $\mathcal{G}$, randomly pick a signal type from "harmonic", "pseudo periodic" and "autoregressive" and generate a time series with length $T$ according to this type. We use the Python library "TimeSynth"[4] to generate such signals. When generating these signals, the stop time is set to 100. For "harmonic", the frequency and the noise std are uniformly drawn from $[0.1, 1.0]$ and $[0.1, 0.3]$, respectively. For "pseudo periodic", the frequency is uniformly drawn from $[1.0, 6.0]$, "freqSD" and "ampSD" are set to 0.0 and 0.1. For "autoregressive", "ar_param" is uniformly sampled from $[0.3, 1.0]$ and "sigma" is uniformly sampled from $[0.01, 0.1]$.

---

[3]https://github.com/cmu-phil/tetrad
[4]https://github.com/TimeSynth/TimeSynth

3. For a variable $x_i$ with parents $\mathbb{PA}(x_i)$ in $\mathcal{G}$, we consider both **linear** relationship $x_i = \sum_{j \in \mathbb{PA}(x_i)} w_j x_j + \epsilon$ and **nonlinear** relationship $x_i = \sum_{j \in \mathbb{PA}(x_i)} w_j \tanh(x_j) + \epsilon$, where $w_j$ is uniformly sampled from $[0.5, 2.0]$ and $\epsilon$ is uniformly sampled from $[-0.1, 0.1]$. The time series for those variables are generated in a topological order.

4. Add anomalies into the generated time series: We consider three types of anomalies. The first one is a "**measurement**" anomaly, i.e., randomly pick a variable $x_i$, a time step $t$, a scale $s$ (uniformly sampled from $[0, 3]$) and a duration $d$ (uniformly sampled from $[5, 20]$), and then set $x_i(t : t + d) = [x_i(t) - \text{median}(x_i)] * s + \text{median}(x_i)$. The second one is an "**intervention**" anomaly, i.e., after generating "measurement" anomalies for some variables, those anomaly values propagate to the children according to the causal mechanisms. The third one is an "**effect**" anomaly where anomalies only happen on the variables with no causal children.

Figure 3 shows the generated causal graph which contains 15 variables. According to this causal graph, we can generate multivariate time series by following the procedure mentioned above, as shown in Figure 4 and Figure 5 where some of the abnormal time steps in the test dataset are labeled by blue triangles.

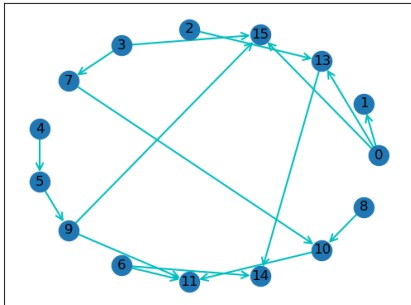

Figure 3: An example of the ground truth causal graph in the simulated datasets.

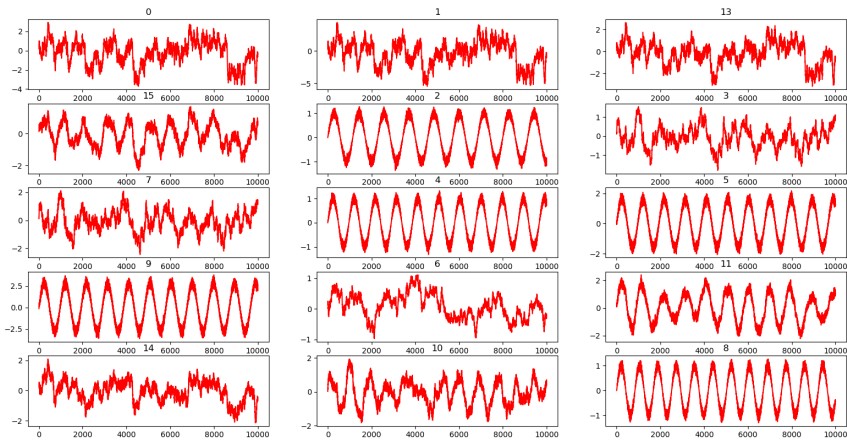

Figure 4: An example of the simulated training datasets.

## A.4 MIXED RELATIONSHIPS: LINEAR + NONLINEAR

We also consider a mix of linear and nonlinear relationships, i.e., randomly pick a relationship from [linear, nonlinear] during time series generation, where the probability of selecting "linear" is $0.7$.

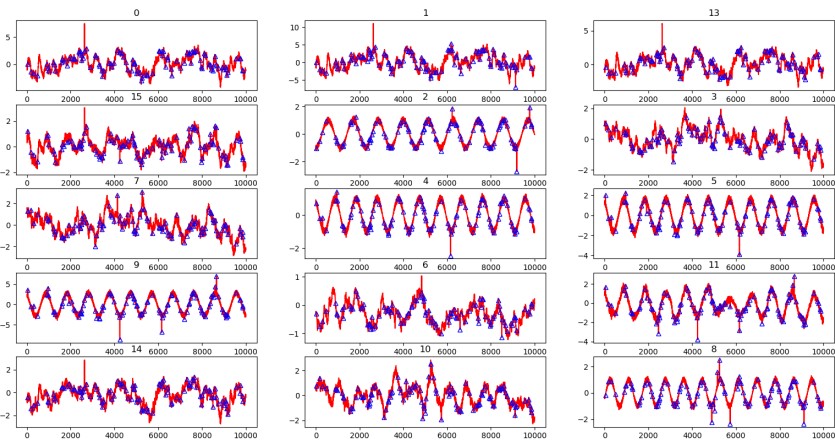

Figure 5: An example of the simulated test datasets. Some of the abnormal time steps are labeled by blue triangles.

Table 8 shows the experimental results when the relationships are a mix of linear and nonlinear. Compared with the linear case, a mixed relationship makes the anomaly detection problem harder, e.g., the performance of most approaches drops. Our approach still has the best performance compared to the other methods.

Table 8: Performance comparison with the simulated dataset. The relationships are a mix of linear and nonlinear.

| Methods | Mix, Measurement | | | Mix, Intervention | | | Mix, Effect | | |
|---|---|---|---|---|---|---|---|---|---|
| | Precision | Recall | F1 | Precision | Recall | F1 | Precision | Recall | F1 |
| IF | 0.401 | 0.542 | 0.461 | 0.241 | 0.618 | 0.347 | 0.223 | 0.329 | 0.266 |
| AE | 0.314 | 0.669 | 0.427 | 0.277 | 0.406 | 0.329 | 0.229 | 0.436 | 0.301 |
| VAE | 0.364 | 0.479 | 0.414 | 0.241 | 0.621 | 0.347 | 0.181 | 0.440 | 0.256 |
| LSTM-VAE | 0.519 | 0.595 | 0.555 | 0.425 | 0.522 | 0.468 | 0.230 | 0.560 | 0.326 |
| DAGMM | 0.603 | 0.601 | 0.602 | 0.638 | 0.494 | 0.557 | 0.619 | 0.582 | 0.600 |
| USAD | 0.252 | 0.454 | 0.324 | 0.217 | 0.315 | 0.257 | 0.154 | 0.467 | 0.231 |
| Ours | 0.751 | 0.720 | **0.735** | 0.635 | 0.794 | **0.706** | 0.805 | 0.591 | **0.682** |

## A.5 MIXED DATA TYPES: DISCRETE + CONTINUOUS

This experiment considers the case that the generated multivariate time series contains both discrete and continuous values. The generation procedure for this kind of time series data has the following differences. For the variables without parents in $\mathcal{G}$, we randomly pick a signal type from "harmoni", "pseudoperiodic" and "autoregressive". For a variable $x_i$ with parents $\mathcal{P}(x_i)$ in $\mathcal{G}$, we first randomly pick a data type, i.e., choosing "discrete" with probability 0.4 and "continuous" with probability 0.6. For "discrete", $x_i = \mathbf{1}[\sum_{j \in \mathcal{P}(x_i)} w_j x_j + \epsilon > 0]$, i.e., logistic regression. For "continuous", $x_i = \sum_{j \in \mathcal{P}(x_i)} w_j x_j + \epsilon$, i.e., linear relationship. When generating anomalies for "discrete", we take $x_i(t : t+d) = \text{ceil}[[x_i(t) - \text{median}(x_i)] * s + \text{median}(x_i)]$ for time step $t$, duration $d$ and scale $s$. Figures 6 and 7 give an example of the generated training data and test data in this case.

Table 9 shows the experimental results with the "discrete/continuous" datasets. Compared with the "continuous" case, the performance of all the methods decreases because inferring correlations or doing causal discovery becomes relatively harder with the mixed data types. Our approach still significantly outperforms the other ones in this case.

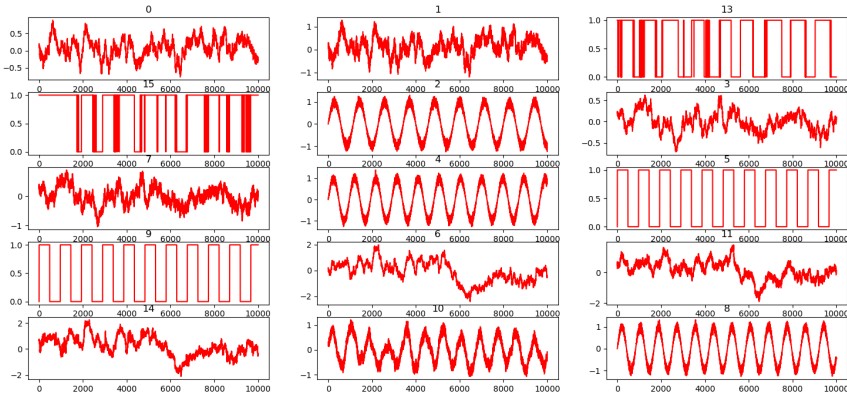

Figure 6: An example of the simulated training datasets with mixed data types.

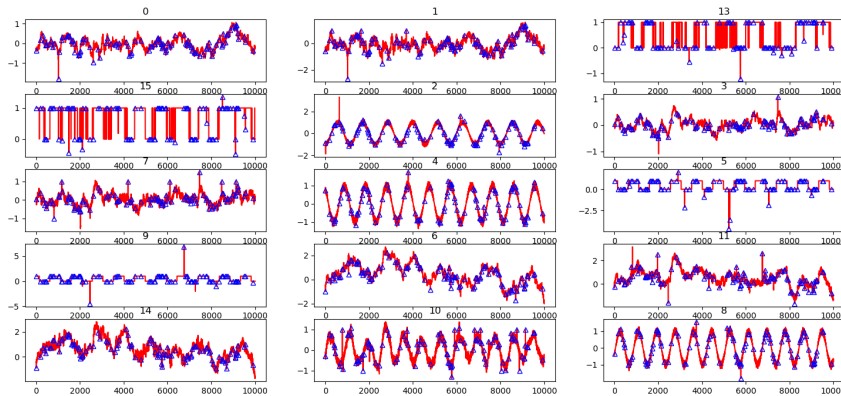

Figure 7: An example of the simulated test datasets with mixed data types. Some of the abnormal time steps are labeled by blue triangles.

Table 9: Performance comparison with the simulated dataset. The data types are a mix of discrete and continuous.

| Methods | DC, Measurement | | | DC, Intervention | | | DC, Effect | | |
|---|---|---|---|---|---|---|---|---|---|
| | Precision | Recall | F1 | Precision | Recall | F1 | Precision | Recall | F1 |
| IF | 0.218 | 0.584 | 0.317 | 0.212 | 0.713 | 0.327 | 0.195 | 0.727 | 0.308 |
| AE | 0.437 | 0.251 | 0.319 | 0.494 | 0.376 | 0.427 | 0.324 | 0.392 | 0.355 |
| VAE | 0.207 | 0.602 | 0.308 | 0.340 | 0.390 | 0.363 | 0.245 | 0.557 | 0.341 |
| LSTM-VAE | 0.291 | 0.484 | 0.363 | 0.365 | 0.633 | 0.463 | 0.282 | 0.547 | 0.372 |
| DAGMM | 0.515 | 0.447 | 0.479 | 0.529 | 0.604 | 0.564 | 0.833 | 0.376 | 0.518 |
| USAD | 0.236 | 0.363 | 0.286 | 0.352 | 0.348 | 0.350 | 0.268 | 0.313 | 0.289 |
| Ours | 0.591 | 0.715 | **0.647** | 0.837 | 0.574 | **0.681** | 0.644 | 0.666 | **0.655** |

## A.6 REAL-WORLD MOTIVATING EXAMPLE

Figure 8 shows why causality matters with a real-world example. In SWaT (Mathur & Tippenhauer, 2016), at timestamp 491, our causality-based approach detects a true anomaly where the causal mechanism between Metrics 1, 0 and 9 is violated (Metrics 0 and 9 are the causal parents of Metric 1). We plot the probability density of the reconstruction error based on the causal mechanism (top left figure), where the black triangle is the anomaly. Clearly, this anomaly can be easily identified w.r.t. the p-value. But if we check the probability density of the "joint" reconstruction error by AE

(top right figure), this anomaly cannot be found w.r.t. the p-value. The bottom figure plots the time series data of these three metrics. Intuitively, we can observe that Metric 1 has a peak value when Metric 0 is low, and Metric 1 is low when Metric 0 is high. In the range (450, 550), this causal mechanism is violated, e.g., Metric 0 is high while Metric 1 is also high. This type of anomalies is hard to be identified by checking joint or marginal distributions.

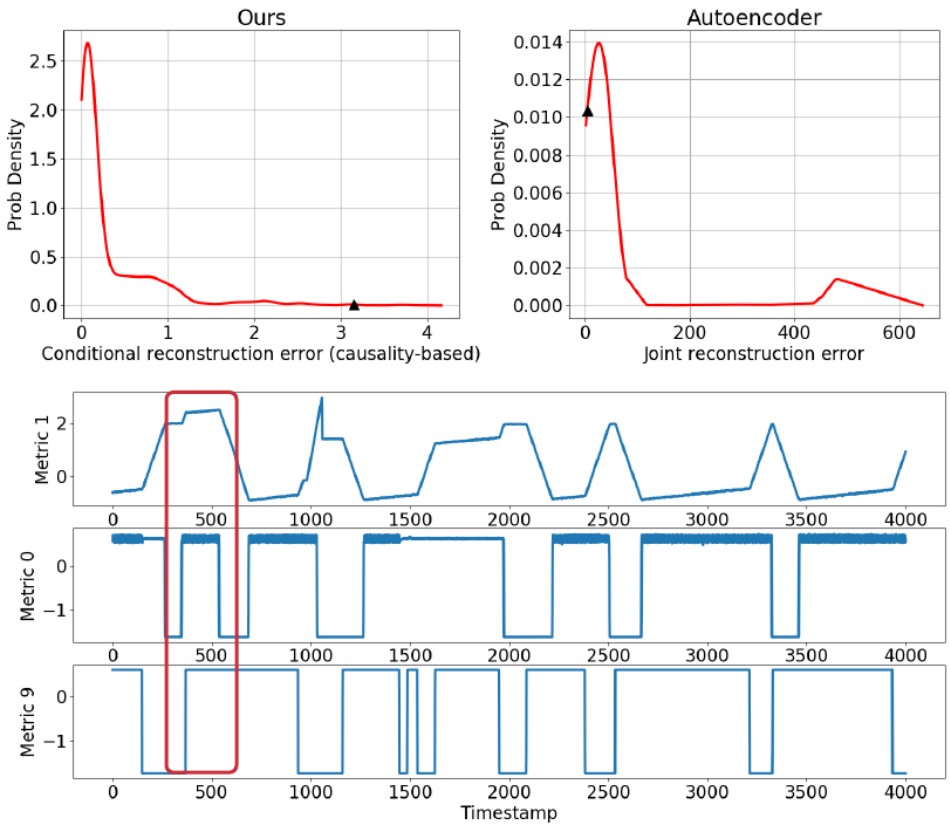

Figure 8: A motivation example in the real-world dataset SWaT (Mathur & Tippenhauer, 2016). At timestamp 491, our causality-based approach detects a true anomaly where the causal mechanism between metrics 1, 0 and 9 is violated (metrics 0 and 9 are the causal parents of metric 1).

## A.7 TRAINING ANOMALIES

When the fraction of anomalous points is large in the training data, these anomalies may decrease detection performance since the discovered causal graph may not be accurate. In this case, we can apply the solution discussed in Section 2.2.4, updating the causal graph and anomaly detection model iteratively. In this experiment, the training and test data are generated under the setting "linear/measurement", and a large proportion of noises are added into the training data, i.e., adding additional Gaussian noises to the first 20% data points in the training data. These noisy data points makes estimating accurate causal graphs harder via causal discovery algorithms. In each iteration, 3% data points are detected as anomalies and removed. Figure 9(a) shows the detection performance on the test dataset measured by the F1 scores over each iteration. In the beginning the discovered causal graph has more errors due to the noises in the training data, leading to the low F1 score. After each iteration, our approach removes the detected anomalies from the training data, making the discovered causal graph more accurate in the next iteration, so that the detection performance increases consistently. This experiment empirically verifies our "iterative updates" approach in the case where the training data has a large portion of anomalies. Figure 9(b) plots the difference between the adjacency matrices of two consecutive estimated causal graphs, which increases first then decreases and converges to 0 since the distribution of training data gradually changes from a

mix of noises and regular points to regular points only. This experiment considers an extreme case

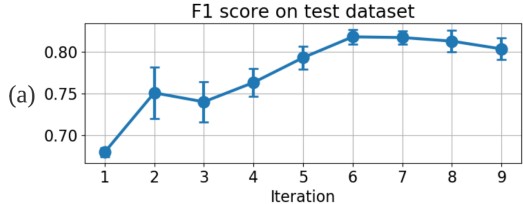 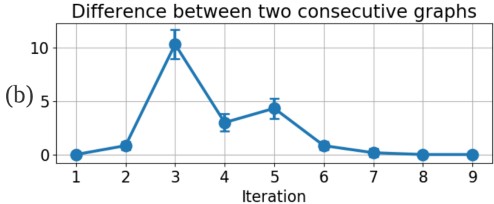

Figure 9: Empirical study on our "iterative updates" approach discussed in Section 2.2.4 for handling large noise in the training data. (a) The detection performance (F1 scores) on the test data over iterations. (b) The difference between the adjacency matrices of two consecutive discovered causal graphs.

that the proportion of anomalies and the magnitude of anomalies are large. In practical applications where the proportion of anomalies in training data is relatively small., e.g., the public datasets, there is no need to apply this iterative approach, i.e., one iteration is good enough.

## A.8  RUNNING TIME

Table 10 shows the running time of our approach. The most time consuming step is local causal mechanism estimation (conditional distribution estimation). After training, our approach detects anomalies and root causes fast.

Table 10: The running time of our approach (wall clock time).

| Stage | SWaT | WADI |
|---|---|---|
| Training (Causal discovery) | 10.27s | 42.18s |
| Training (Conditional distribution estimation) | 415.37s | 1026.59s |
| Inference (anomaly detection) | 0.279ms per point | 0.636ms per point |

## A.9  ABLATION STUDY ON CAUSAL GRAPH $\mathcal{G}$

We also studied the effects of different parameters for discovering causal graphs on the performance of our approach. The parameters that we investigated are "max degree" and "penalty discount" in FGES, both of which affect the structure of the causal graph, e.g., sparsity, indegree, outdegree. In this experiment, we use 6 different "max degree" $[5, 6, 7, 8, 9, 10]$ and 6 different "penalty discount" $[20, 40, 60, 80, 100, 120]$. Smaller "max degree" or larger "penalty discount" leads to more sparse graphs with less edges, e.g., for SWaT, the number of the edges in $\mathcal{G}$ is $[70, 79, 88, 95, 98, 102]$ when "max degree" $= [5, 6, 7, 8, 9, 10]$, respectively.

Figure 10 plots the detection precision, recall and F1 score obtained with different "max degree" and "penalty discount". For SWaT, these two parameters don't affect the performance much. For WADI, when "max degree" decreases (the causal graph becomes more sparse) or "penalty discount" decreases (the causal graph has more false positive links), the performance also decreases but it doesn't drop much, i.e., the worst F1 score is still above 0.65. When "max degree" $> 6$ and "penalty discount" $> 40$, we got similar performance, e.g., the F1 score is around 0.8, showing that our approach is robust to the changes of the inferred causal graph. In practice, the causal graph is not required to be accurate, namely, we just need to ensure that it doesn't contain too many missing links or false positive links.

## A.10  DETECTED ANOMALIES IN PUBLIC DATASETS

Figures 11-14 show the detection results by our approach where we did downsampling for better demonstration. The left figures plot the ground truth labels. The right figures plot the detected anomalies in a point-adjust way.

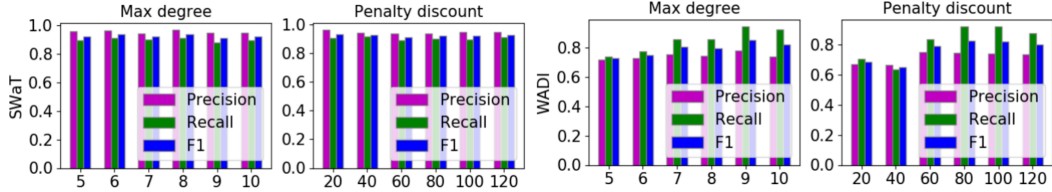

Figure 10: Precision, Recall and F1 score as a function of "max degree" and "penalty discount". The first and second rows plot the metrics for SWaT and WADI, respectively.

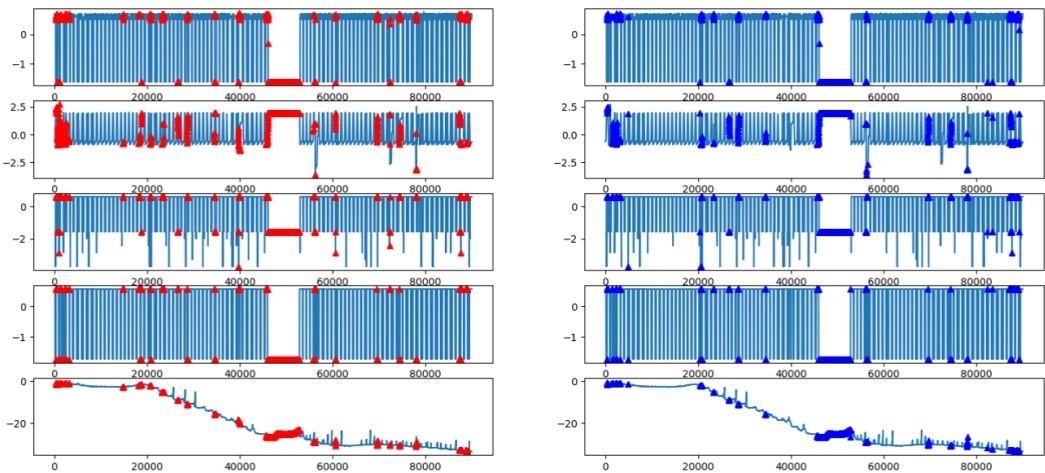

Figure 11: SWaT detection results. Left: Ground-truth labels. Right: Detected anomalies.

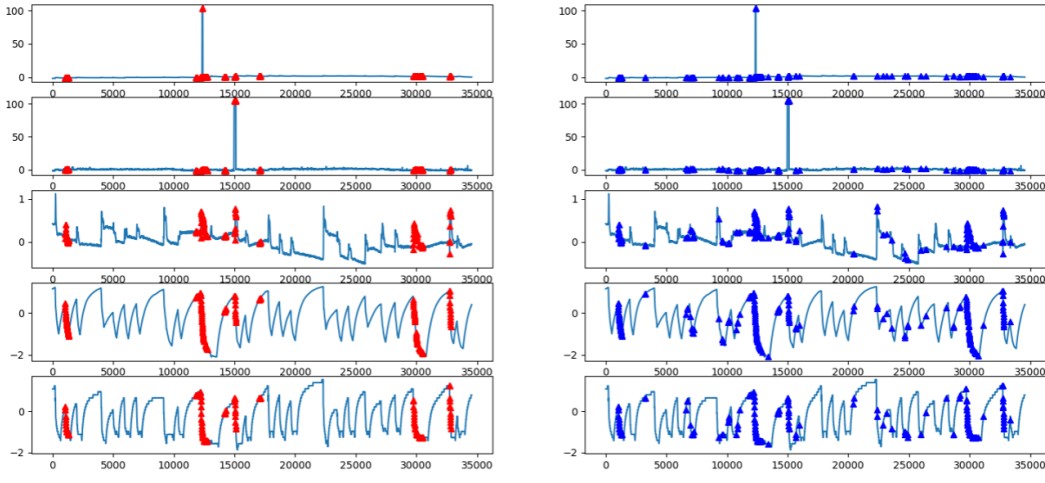

Figure 12: WADI detection results. Left: Ground-truth labels. Right: Detected anomalies.

## B    CASE STUDY: REAL-WORLD APPLICATIONS IN AIOPS

Root Cause Analysis (RCA) in real-world applications such as AIOps can be very challenging. One practical issue for identifying root causes is that an anomaly occurs in a parent often makes its contemporaneous causal children abnormal due to the estimation errors in conditional distributions. To handle this issue, we developed the following practical algorithm.

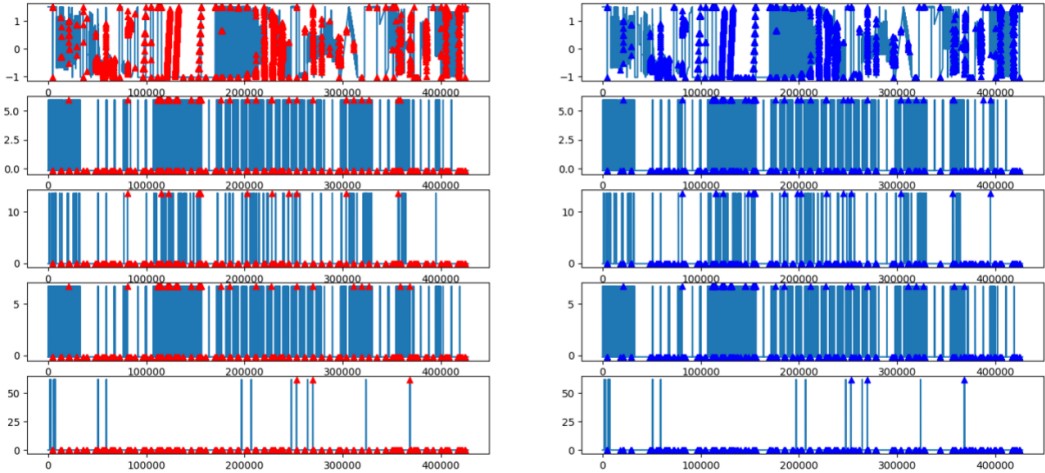

Figure 13: SMAP detection results. Left: Ground-truth labels. Right: Detected anomalies.

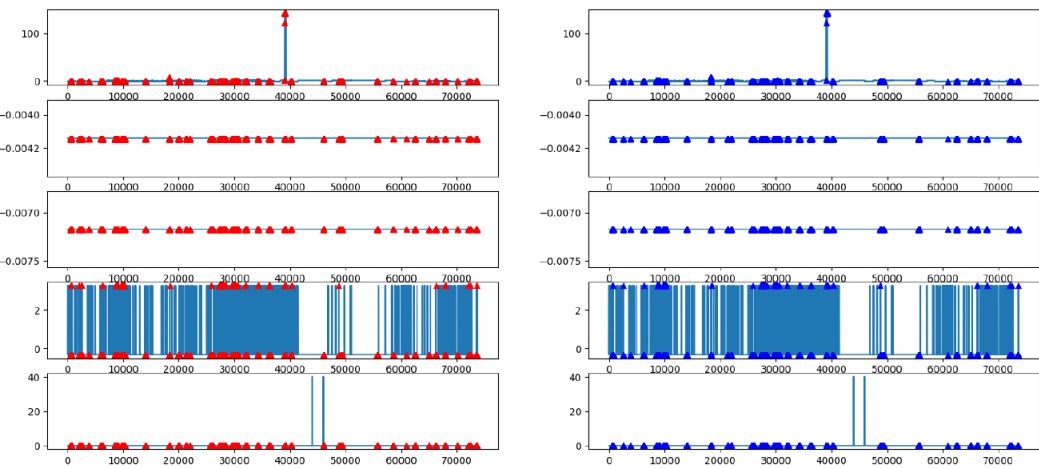

Figure 14: MSL detection results. Left: Ground-truth labels. Right: Detected anomalies.

Specifically, for a variable $x_i$, recall that its root cause score at time $t$ is

$$\mathbb{S}(x_i(t)) = 1 - \mathbb{M}_i(x_i(t)). \tag{2}$$

Suppose that $\mathcal{N}(x_i(t))$ is the set of the contemporaneous causal children of $x_i(t)$, the final root cause score is define by

$$\mathbb{RS}(x_i(t)) = \mathbb{S}(x_i(t)) + \alpha \frac{1}{|\mathcal{N}(x_i)|} \sum_{x_j(t) \in \mathcal{N}(x_i)} \mathbb{RS}(x_j(t)), \ \forall i = 1, \cdots, d, \tag{3}$$

where $\alpha$ is a weight parameter satisfying $0 \le \alpha < 1$. When $\mathcal{N}(x_i)$ is empty, we set $\mathbb{RS}(x_i(t)) = \mathbb{S}(x_i(t))$. Here the final root cause score of a variable is the combination between its original root cause score and the root cause scores of its children. The final root cause scores for all the variables can be computed by these linear equations. When $\alpha = 0$, it is reduced to the ideal scenario discussed above. When $\alpha \ne 0$, the above approach improves the ranking of root causes from a global view. Then the root causes at time $t$ can be identified by picking the variables with top root cause scores.

Figure 15 shows another major incidence. The top abnormal variables are SYIO (system I/O), USIO (user I/O), Lfpw (log file parallel write), UTIL (I/O utilization). All of them are related to I/O issues, meaning that the root causes are the components related to I/O.

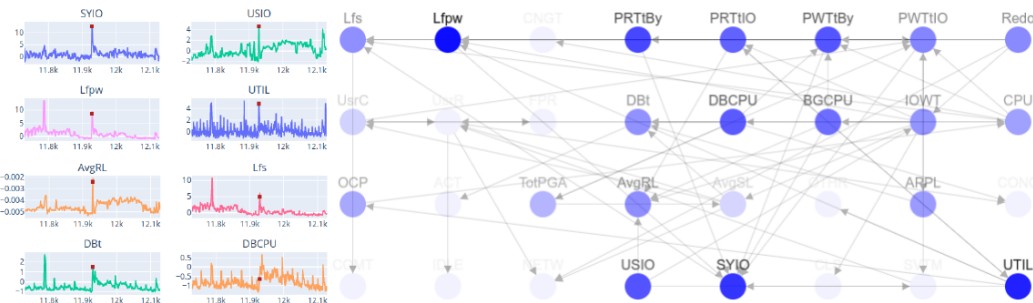

Figure 15: Case study in AIOps. 8 out of 61 time variables (left) and a part of the causal graph (right). The anomaly scores are indicated by the colors, e.g., deeper colors indicate larger anomaly scores.

