# OpenReview forum: "A Causal Approach to Detecting Multivariate Time-series Anomalies and Root Causes"
_ICLR.cc/2023/Conference — Submitted to ICLR 2023_

### Official Review · Reviewer_oDPS · 2022-10-24

**Confidence:** 3
**Correctness:** 3
**Technical Novelty And Significance:** 3
**Empirical Novelty And Significance:** 3
**Recommendation:** 6

**Clarity, Quality, Novelty And Reproducibility:**

The paper is well-written and easy to follow. In terms of novelty, approaching the anomaly detection problem from the causal perspective seems new to me, although the methods used to identify the causal structure are somewhat standard. The paper did not include many details about the implementation and hyperparameter settings, which could limit the reproducibility of the paper. I would like to encourage the authors to discuss more details about the implementation and the hyperparameter setting (e.g., $\lambda$) and release the source codes to maximize reproducibility.

**Strength And Weaknesses:**

Strengths:
- Modeling the anomaly detection problem from the causal perspective is quite interesting and important. The detection of root causes also makes the model more explainable.
- The performance obtained using multiple datasets is very impressive.

Weaknesses:
- I am not fully convinced by the assumption that the causal structure is a DAG. First, the DAG may not be able to capture the time lag of the causal relationship. Second, there could be positive/negative feedback loops over time. For example, in economics, lowering investment causes lowering income after a certain lag, and in turn, causes lowering investment after some lags. In the summary causal graph, there could be cycles. Could some patterns be captured and modeled as well?
- It seems that the detection algorithm relies heavily on the preset threshold $\lambda$. It is not clear how much it impacts the performance and how the users should select the best hyperparameter.

**Summary Of The Paper:**

This paper focuses on anomaly detection from time series. Specifically, this paper tackles this problem from a causal perspective. It first learns a causal structure from the data. The causal structure is assumed to be a directed acyclic graph (DAG) which captures stationary causal structure. The anomaly then can be defined as data points that do not follow the probability conditional on its causal parents. By learning the causal structure, the root cause of the anomaly can also be detected. Empirical evaluation with both simulation and public datasets shows that the proposed method obtains impressive results.

**Summary Of The Review:**

Overall, this paper offers a new perspective to tackle the anomaly detection problem for time series data. The idea is reasonable. The proposed method is overall sound, although some parts of the methodological design need stronger justification and more explanations. The empirical evaluation shows that the proposed method is impressive.

---

> ### Author Response · Authors · 2022-11-17
> **We explain the DAG assumption and discuss how to set detection thresholds below**
>
> 1. I am not fully convinced by the assumption that the causal structure is a DAG.
>
> Time-lagging causal relationships can also be described in a DAG. For example, let's consider a causal relationship X(t-1) -> Y(t-1) -> X(t). We take X(t-1) and X(t) as separate nodes in the causal graph, so that this time-lagging relationship can be represented as a DAG. Therefore, if the granularity of the time series is high, i.e., time-lagging causal relationships can be discovered, a DAG is enough to represent all the time-lagging or simultaneous casual relationships. If the granularity is low, e.g., the difference between t-1 and t cannot be distinguished, there may be cycles. In this paper, we assume that the granularity is high enough for these time-lagging causal relationships, so that we assume the causal structure is a DAG.
>
> 2. It seems that the detection algorithm relies heavily on the preset threshold. It is not clear how much it impacts the performance and how the users should select the best hyperparameter.
>
> Almost all anomaly detection methods require setting a detection threshold for raising anomaly alerts. If the threshold is set too low, we will get more false-positive alerts. If the threshold is set too high, we will miss more important (false-negative) alerts. In practice, if the training dataset has anomaly labels, the threshold can be set by maximizing precision, recall or F1-score based on different requirements. If there are no anomaly labels, the threshold is usually set to the n-percentile of the training anomaly scores, where n is selected based on domain knowledge. After the threshold is set, we also need to tune it based on real-world performance if the solution is deployed in the application.

---

### Official Review · Reviewer_UaBN · 2022-10-26

**Confidence:** 3
**Correctness:** 4
**Technical Novelty And Significance:** 2
**Empirical Novelty And Significance:** 3
**Recommendation:** 5

**Clarity, Quality, Novelty And Reproducibility:**

The paper's writing is clear. The authors use the proper technical language.

I need the authors' response to the paper mentioned in the previous section to finalize my evaluation of the novelty.

**Strength And Weaknesses:**

### Strengths
1. The idea is simple, makes sense, and should work.
2. The paper is well-written and easy to follow.

### Weaknesses
1. The authors seem to miss a similar paper. Can you describe the relationship of the paper to the idea of the following paper?
    * Budhathoki, K., Minorics, L., Blöbaum, P., & Janzing, D. (2022). Causal structure-based root cause analysis of outliers. In _International Conference on Machine Learning_.
2. The authors should comment on the error propagation from the DAG learning step. They have a set of ablation experiments on this topic, but might not be convincing.

**Summary Of The Paper:**

The authors propose a two-step procedure to perform root-cause analysis (RCA):
1. Learn a causal graph from the data,
2. Identify the root causes based on the changes in the local mechanisms (conditional probabilities).

The paper is simple and easy to read. The experiments look extensive. There are doubts about the novelty of the idea though.

**Summary Of The Review:**

The authors propose a two-step procedure to perform root-cause analysis (RCA):
1. Learn a causal graph from the data,
2. Identify the root causes based on the changes in the local mechanisms (conditional probabilities).

The paper is simple and easy to read. The experiments look extensive. There are doubts about the novelty of the idea though.

---

> ### Author Response · Authors · 2022-11-17
> **The comparison with the mentioned paper and the discussion about DAG**
>
> 1. The authors seem to miss a similar paper. Can you describe the relationship of the paper to the idea of the following paper?
>
> The paper mentioned by the reviewer only considers the root cause analysis problem, i.e., understanding detected anomalies or outliers, which cannot be used for anomaly detection. Besides, it considers counterfactuals for root cause analysis, i.e., given a functional causal model, it finds the root cause of a detected anomaly/outlier by computing the contribution of each noise term to the anomaly score, where the contributions are symmetrized using the concept of Shapley values.
>
> Our paper presents a causality-based framework for both anomaly detection and root cause analysis, allowing generating anomaly alerts and the corresponding root causes simultaneously.
> Our approach doesn't consider counterfactuals. We define anomalies and root causes directly based on the modularity property from the causal perspective (Definition 2). Therefore, our approach is totally different from their paper, although both approaches utilize causal structures. We also discussed how to discover causal structures and how to model causal mechanisms.
>
> 2. The authors should comment on the error propagation from the DAG learning step. They have a set of ablation experiments on this topic, but might not be convincing.
>
> For real-world datasets, the discovered causal graph always has errors due to data issues, e.g., data is noisy, or the assumptions of causal discovery algorithms are violated. In the experiments, we showed that the errors in the causal graph do not affect anomaly detection performance much, e.g., Table 6 and Figure 10. But for root cause analysis, a more accurate causal graph is required. If the causal relationships are incorrect, the identified root causes can also be wrong. Therefore, as shown in our AIOps application, we combine casual discovery and domain knowledge to generate a more reliable causal graph. We also discussed the discovered causal graph with our engineering team for reducing errors as much as possible.

---

### Official Review · Reviewer_FAmL · 2022-10-27

**Confidence:** 5
**Correctness:** 2
**Technical Novelty And Significance:** 3
**Empirical Novelty And Significance:** 2
**Recommendation:** 3

**Clarity, Quality, Novelty And Reproducibility:**

The paper is well written and easy to understand. It's lucidity reveals that many of the concepts introduced may not be well understood by the authors.  It would help if the paper would pick a consistent theoretical basis, as the Bayesian probabilistic nature of the methods used for causal discovery (e.g. Pearl's work) suggest.  WIthout revealing why the proposed method works  questions about reproducibility cannot be determined.

As for comparisons with previous work, there are a wide range of vector anomaly detection methods based on PCA.  One that is in widespread use that must be considered is:
A. Lakhina, M. Crovella, C. Diot, “Diagnosing Network-Wide Traffic Anomalies”
SIGCOMM’04, Aug. 30–Sept. 3, 2004, Portland, Oregon, USA.


**Strength And Weaknesses:**

The fundamental notion addressed by this paper is sound, and the conceptual approach is well thought out. This is a novel application that showcases the wide applicability of causal reasoning, both for discovering properties in data, and for understanding the real-world phenomena that the data represent. I tend to believe that this approach is better than the current methods.

However the the paper does not elucidate a coherent method that fits into a mathematical model. What I see is a jumble of ideas, inconsistent and inaccurately presented. There are an overwhelming number of misperceptions that do not recommend the work:

- p.2 p(z) = 1.695  THis is not a probability. Probabilities are bounded between 0 and 1
-  p(y|x) cannot be considered a "p -value".  A p-value is a kind of tail probability of a statistical test.
- In the probability p(y|x) x is the cause. Conventionally the influence goes from x, the cause, to y, the effect.
- Why use a p-value test [ Heard & Rubin-Delanchy (2018)] when you have likelihoods, the proper Bayesian solution to this problem? This avoids the need to set a threshold on a value -- the likelihood of the anomaly can be reported instead.
-The structure learning methods cited, although informally referred to as "causal discovery" methods are just structure learning ("Markov factorization") methods, and cannot generate causal claims without additional assumptions, or domain input.
- The authors propose several ways to treat of temporal dependencies within one time-series, e.g. such as ARMA, without realizing that these dependencies fit into the structure learning paradigm used between timeseries. The autocorrelation within a time-series is just another dependency that, to make a consistent, comprehensive model should be incorporated into the learned network.  Then the method could detect changepoints -- something the current method appears to overlook.



**Summary Of The Paper:**

This paper is based on the insight that anomaly detection can be treated as a causal discovery problem, to both improve detection and to facilitate root cause analysis. An anomaly is defined as a datum that does not fit the discovered data-generating process, in short an instance where the process is non-stationary. The graph decomposition of the timeseries by structure learning has heuristic value to pin-point the variable that causes the anomaly. The proposed method is demonstrated on simulation data and an extensive set of public datasets.


**Summary Of The Review:**

The paper is a poorly developed approach to a good idea, that cannot support the empirical claims made by the paper. Without a better understanding, offered by a consistent mathematical argument why the results were obtained, there remain too many questions about the results, and detract from their credibility.

---

> ### Author Response · Authors · 2022-11-17
> **We believe there are some misunderstandings about the problem formulation and our causality-based approach**
>
> We believe there are some misunderstandings about the problem formulation and our causality-based approach. We will appreciate it if you can check our responses.
>
> We view anomalies in multivariate time series as instances that do not follow the regular causal mechanisms, and propose a causality-based framework for both anomaly detection and root cause analysis. Although root cause analysis is a harder problem than anomaly detection, our approach can detect anomalies and root causes simultaneously based on our causal perspective. Our approach doesn't treat anomaly detection as a causal discovery problem, i.e., causal discovery is only one of the steps. Our experiments also show that our causality-based approach boosts the anomaly detection and RCA performance a lot compared with previous approaches on either simulation datasets or public datasets, demonstrating the effectiveness of our approach.
>
> For the concern about the mathematical model, let's consider a multivariate time series X (with d variables), and denote the joint distribution of X_t (at timestamp t) as P(X_t). If we consider time-lagging dependencies, the joint distribution of X_t is P(X_t | X_{t-1}, X_{t-2}, ...). For simplicity, we use P as the joint distribution. The time series anomaly detection problem considered in this paper is to test whether an observed data X_T follows its regular distribution P, i.e., X_T is labeled as an anomaly if it does not follow P.
>
> The joint distribution P can be factorized into local causal mechanisms as discussed in Page 3. Therefore, X_T is an anomaly if there exists at least one variable x_i in X_T that violates its local causal mechanism (Definition 1), and the root causes can be naturally defined (Definition 2).
>
> The time series anomaly detection problem then becomes: Given the local causal mechanisms (estimated from data in practice, Sections 2.2.1 and 2.2.2), for an observed data X_T, test whether there exists a variable in X_T that violated its local causal mechanism modeled by conditional probability distribution. Therefore, the problem involves n statistical tests where n is the number of local causal mechanisms. That's why we consider combining p-values from individual tests as discussed in Page 4.
>
> Other questions:
>
> 1. p.2 p(z) = 1.695 This is not a probability. Probabilities are bounded between 0 and 1.
>
> Here, "p" represents **probability density** (we have highlighted it in Page 2).
>
> 2. p(y|x) cannot be considered a "p-value". A p-value is a kind of tail probability of a statistical test.
>
> "P-value" mentioned in Figure 1 is not p(y|x). Anomaly detection considers a statistical test, i.e., whether an observed data point follows its regular distribution. If it doesn't follow, it is treated as an anomaly or outlier. So "p-value" here is used for this statistical test.
>
> 3. Why use a p-value test [Heard & Rubin-Delanchy (2018)] when you have likelihoods, the proper Bayesian solution to this problem? This avoids the need to set a threshold on a value -- the likelihood of the anomaly can be reported instead.
>
> For anomaly detection, the outputs of an anomaly detector should be anomaly labels (0 or 1) instead of anomaly likelihood or scores, because an anomaly detector is used to detect incidents in a real-world system. If the detector only provides anomaly scores, we still cannot determine whether it is an incident or not. Therefore, a threshold is needed to make such a decision. A threshold can be determined in many ways, e.g., based on past experience, domain knowledge, or statistical analysis. Here a "p-value" test is applied to compute this threshold. After training, the anomaly detector generates a set of anomaly scores which follow a "regular" distribution (if the training dataset has anomalies, please refer to Section 2.2.4). In the detection phase, given the observed data, the algorithm computes its anomaly score and tests if this score follows the "regular" distribution to determine whether it is an anomaly.
>
> 4. The authors propose several ways to treat of temporal dependencies within one time-series, e.g. such as ARMA, without realizing that these dependencies fit into the structure learning paradigm used between timeseries. The autocorrelation within a time-series is just another dependency that, to make a consistent, comprehensive model should be incorporated into the learned network. Then the method could detect changepoints -- something the current method appears to overlook.
>
> This is a good suggestion. Actually, we also consider including autocorrelation into our framework to handle both anomaly detection and change-point detection problems. Because including autocorrelation makes our framework more complicated and hard to follow, we only focus on anomaly detection and root cause analysis problems in this paper. Our future work will extend this framework to handle change-point detection problems.

---

### Official Review · Reviewer_8A4e · 2022-12-03

**Confidence:** 4
**Correctness:** 3
**Technical Novelty And Significance:** 3
**Empirical Novelty And Significance:** 3
**Recommendation:** 6

**Clarity, Quality, Novelty And Reproducibility:**

The paper is well-written, the quality of the manuscript is high. The experimental procedure is clearly laid out so I am guessing the results could be reproduced.

On the novelty side, using casual models to explain and detect anomalies is interesting.

**Strength And Weaknesses:**

Strengths:

- Novel framework which can be enhanced with advances in causal discovery methods.
- Authors provide the required background of causal inference, making this work accessible to readers new to the domain.
- The manuscript is well-written with minimal to no clerical errors, accompanied with simple examples and plots which makes it a moderately easy read for non-experts. The structure of the paper is good and easy to follow.

Weaknesses:

- Method highly sensitive to causal graph discovery method, which in itself is sensitive to the quality of data (which in this case are anomalies). Some more discussion on this is needed.
- Proposed approach uses a window of time to extract samples for analysis, this does capture the time dependences between data dimensions however, it cannot capture the temporal dimension beyond the window of time chosen.
- There are some limitations related to the size of the window and the underlying assumption that the causality is localized within that window. What happens if the data has periodicity? What would be the appropriate window size in that case?

**Summary Of The Paper:**

In the present manuscript, author(s) offer a causality based solution to anomaly detection in multivariate time series data. Such data can be ubiquitously found in real world scenarios and one might be interested in finding any anomaly in this data and its cause. Existing set of solutions either perform separate univariate anomaly detection and ignore the dependences between the various dimensions or a direct multivariate anomaly detection which does capture the inter-dimensional dependencies however is still incapable of learning the underlying data generating process and thus require a separate root cause analysis for anomaly cause detection. The present work attempts to detect anomalies and their cause simultaneously while capturing dependencies between the dimensions of the time series data by factorizing the joint distribution (data) into simpler, local mechanisms which are modular.

On a high level, the proposed framework, from the training data first generates a causal graph using data appropriate choice of causal graph discovery algorithms (e.g. PC, FCI, GES, etc). The inferred causal graph is then used to factorize the joint distribution using the Markov factorization. These factorizations are conditional probabilities corresponding to local mechanisms, which can be computed using methods such as kernel density estimation. Now for the non-training input (multivariate time series data), the anomaly score is computed as an aggregation of individual p-values, one for each variable and its corresponding mechanism. Anomaly score greater than a certain threshold is considered as anomaly and the variable with highest root cause score is deemed the cause of anomaly. Authors follow existing approaches from Heard & Rubin-Delanchy (2018) compute the anomaly score and ranking based algorithm for root cause score.

The experiments are thorough but not exhaustive. Experiments with more complex data generating processes are not explored, which is where the causal graph discovery methods are put to test.


**Summary Of The Review:**

This is a well-written paper with some clear advantages in terms of explainability and modeling of anomalies. However, there are limitations in terms of the size of the window, which are not discussed in the paper.

---

### Author Response · Authors · 2022-11-17
**Highlights and changes**

The key novelty:

Instead of treating anomaly detection and root cause analysis (RCA) as two separate stages/problems, we consider anomaly detection and RCA as one problem from a causal perspective, i.e., view anomalies as instances that do not follow the regular causal mechanisms, so that the anomalies and the corresponding root causes can be identified simultaneously. We did an extensive experimental study of our approach and showed that it achieves much better anomaly detection and RCA results than the other compared approaches.

Common strengths mentioned by the reviewers:

The proposed causality-based framework is interesting, important and makes sense. The experiments on both simulation and public datasets show the proposed approach obtains impressive results.

Common questions raised by the reviewers:

1. The DAG assumption: A DAG can model both time-lagging causal relationships and simultaneous causal relationships. For time-lagging causal relationships, the lagged variables are also treated as nodes in the causal graph. For instance, Y_t = aX_t + bX_{t-1} can be represented as X_t -> Y_t <- X_{t-1}. As mentioned by Reviewer FAmL, our framework can also include autocorrelation relationships.

2. How to set the detection threshold: The goal of anomaly detection is to determine whether an observed data point is an anomaly, so the detection threshold is necessary to convert an anomaly score into an anomaly label, e.g., 0 or 1. If the training dataset contains anomaly labels, the detection threshold can be optimized by maximizing detection accuracy. But in most real-world use cases, it is costly to collect a labeled dataset or very few anomaly labels are available. In this case, the detection threshold is determined manually by setting a proper threshold, and then optimized or tuned based on real performance.

Paper changes:

1. We revised the problem definition in Section 2 and revised the caption in Figure 1 to resolve misunderstandings on "p-value".
2. We highlighted the key contribution of this paper, i.e., a causality-based framework for detecting anomalies and corresponding root causes simultaneously.
2. The paper "Causal structure-based root cause analysis of outliers" is discussed in the related work section.
3. How to set the detection threshold is highlighted in the appendix.

---

### Decision · Program_Chairs · 2023-01-20

**Decision:**

Reject

**Justification For Why Not Higher Score:**

The paper does not clearly explain why the method achieves the high performance. The development of the method is enumerated. Some analysis should be added or helpful insight should be provided.

**Justification For Why Not Lower Score:**

N/A

**Metareview: Summary, Strengths And Weaknesses:**

This paper presents a method for detection anomalies and the corresponding root causes simultaneously in multivariate time series. The topic itself is timely and important, which can be applied to many real-world systems. The underlying mechanism is represented by a causal graph, which is found by off-the-shelf causal discovery algorithms. Anomaly and root cause scores are calculated based on deviations from local causal mechanisms that are learned from the training data. The paper is well written and the idea is simple and sound. The validity of the proposed method is justified by experiments. There are mixed reviews with a few concerns. Even after the authors’ rebuttal, all reviewers stood by their decision unfortunately. Certainly, there is some misunderstanding for a particular reviewer, but it is a minor problem. Without any doubt, I believe the paper presents an interesting and important approach to anomaly detection. However, I would like to suggest a few things to improve the paper for future submissions. Learning a causal graph from noisy data is not perfect. Experiments demonstrate that errors in a causal graph do not affect the anomaly detection performance. Experiments are important, but do not tell everything. Thus, it would be nice to add helpful insight or analysis to explain why it happens. In addition, why the method achieves the high performance compared to baseline methods is not clear, except for numerical experiments. It would be also nice to improve the experiments on detecting root causes. For instance, comparing it with [Budhathoki, 2022] would be good. Therefore, the paper is not recommended for acceptance in its current form. I hope authors found the review comments informative and can improve their paper by addressing these carefully in future submissions.